# Technical note: Including non-evaporative fluxes enhances the accuracy of isotope-based soil evaporation estimates

Han Fu<sup>1</sup>, Ming Gao<sup>1</sup>, Huijie Li<sup>2</sup>, Daniele Penna<sup>3,4</sup>, Junming Liu<sup>2</sup>, Bingcheng Si<sup>2,5</sup>, and Wenxiu Zou<sup>1</sup>

- <sup>1</sup> State Key Laboratory of Black Soils Conservation and Utilization, Northeast Institute of Geography and Agroecology, Chinese Academy of Sciences, Harbin 150081, China
  - <sup>2</sup> College of Hydraulic and Civil Engineering, Ludong University, Yantai 264025, China
  - <sup>3</sup> Department of Agriculture, Food, Environment and Forestry, University of Florence, Florence, Italy
  - <sup>4</sup> Forest Engineering Resources and Management Department, Oregon State University, Corvallis, USA
  - <sup>5</sup> Department of Soil Science, University of Saskatchewan, Saskatoon, SK S7N 5A8, Canada
- 10 Correspondence to: Wenxiu Zou (zouwenxiu@iga.ac.cn)

Abstract. Accurately estimating soil water evaporation is essential for quantifying terrestrial water and energy. Isotope-based methods are useful but often rely on steady-state (SS) soil water storage assumptions or non-steady-state (NSS) models that ignore non-evaporative fluxes (such as infiltration and transpiration), leading to mass balance errors. Here, we introduce a new framework, named ISONEVA (ISOtope based soil water evaporation estimation considers dynamic soil water storage and Non-EVAporative fluxes), adapted from lake evaporation models to account for both evaporative and non-evaporative fluxes in soils under dynamic soil water storage. Validation under virtual and field scenarios demonstrated that ISONEVA improved evaporation estimates by 54.1%–83.6% (virtual) and 54.5%–92.4% (field) compared to traditional SS and NSS models. Furthermore, ISONEVA estimated a plausible upper limit of the E/ET ratio (0.139), encompassing the observed value (0.126), whereas SS and NSS methods severely underestimated (0.037) or were unable to produce a limit under field validation. These results highlight the critical role of dynamic soil water storage and non-evaporative fluxes in isotope-based soil water evaporation estimates, offering a robust framework for long-term assessments and informing future coupled land surface modelling efforts.

#### 1 Introduction

Evaporation is a fundamental component of the water and energy balance, consuming nearly one-quarter of incoming solar energy and playing a critical role in land-atmosphere interactions (Or et al., 2013; Trenberth et al., 2009). The long-term (decades) ratio of soil water evaporation (from here onward, simply termed as soil evaporation) to precipitation (E/P) provides key insights into ecohydrological processes, supports accurate water balance assessments, informs evapotranspiration (ET) partitioning, and improves hydrological model calibration (Benettin et al., 2021; Kool et al., 2014; Vereecken et al., 2016).

Stable isotopes in the water molecule (<sup>2</sup>H and <sup>18</sup>O) have emerged as a powerful tool to directly estimate soil evaporation by tracing the enrichment in heavy isotopes (δ<sup>2</sup>H and δ<sup>18</sup>O) in upper soil layers caused by evaporation-driven fractionation (Bailey et al., 2018; Rothfuss et al., 2020). Soil water evaporation and resulting isotope fractionation are highly transient due to dynamic solar radiation, wind speed and other meteorological factors. However, current isotope-based approaches rely on either steady-state (SS) or non-steady-state (NSS) frameworks. SS assumes constant soil water storage and isotopic composition over time, a condition rarely met in dynamic soil systems (Al-Oqaili et al., 2020; Xiang et al., 2021), yet its core assumption of constant water volume is only valid for large water bodies. NSS accounts for temporal variations in storage and isotopes but considers only evaporative fluxes (Gibson and Reid, 2010), neglecting subsurface flow (such as infiltration, root water uptake fluxes, and drainage), which can lead to biased estimates of evaporation (Mattei et al., 2020; Yidana et al., 2016). For example, some studies using NSS methods reported higher evaporation in forest sites compared to shrublands under similar meteorological conditions (Sprenger et al., 2017), contrasting the expectation that shrublands should exhibit greater soil evaporation due to more exposed soil and less canopy cover than forest (Benettin et al., 2021; Nicholls et al., 2023; Nicholls and Carey, 2021; Yu et al., 2022).

This discrepancy may reflect the influence of additional processes not fully accounted for in NSS methods, emphasizing the importance of explicitly representing non-evaporative fluxes, such as percolation and root water uptake, to ensure soil water and isotope mass balance when modelling soil evaporation. To address these limitations, we developed a new framework named ISONEVA (ISOtope based soil water evaporation estimation considers dynamic soil water storage and Non-EVAporative fluxes), extending the formulations originally derived for open water bodies (Gonfiantini, 1986). ISONEVA explicitly incorporates both evaporative and non-evaporative fluxes in the topsoil layer, offering a more realistic representation of soil processes and better soil water and isotope mass balance.

ISONEVA method is evaluated through a combination of virtual test and field lysimeter data, directly comparing it with SS and NSS approaches. By overcoming key theoretical and practical limitations of existing methods, ISONEVA is expected to be a promising tool for advancing soil evaporation assessments in diverse ecosystems and supports improved water resource management under climate changes. This study begins by outlining the theoretical basis of the ISONEVA framework and then evaluates its performance through a combination of virtual and field datasets. The objective is to explore the method's advantages, limitations, and its broader applicability in isotope-based hydrological studies.

### 2 Material and Methods

#### 60 2.1 Method derivatives

A coordinate system is established with the zero-flux plane positioned at the soil surface, and the downward direction defined as positive. Within this framework, fluxes in the topsoil layer include precipitation (P), evaporation (E), and

percolation (Q). P and E have positive and negative directions, respectively; while the direction of Q depends on the balance between P and E: when E exceeds P over a given period, Q can be negative; conversely, when P exceeds E, Q is typically positive (Figure 1). Note that Q can be interpreted more broadly as the sum of all non-evaporative fluxes (do not result in significant isotopic fractionation) that leave the topsoil layer (positive sign), such as percolation and root water uptake (Fu et al., 2025).

Water fluxes:

Isotopic fluxes:

Figure 1. Illustration of soil water and isotopic fluxes within the topsoil layer. P, E, Q are precipitation, evaporation, and percolation flux, respectively.

Based on the defined system, the soil water and isotope mass balance can be written as:

$$\frac{\partial \theta}{\partial t} = -\frac{\partial q}{\partial z} \tag{1}$$

$$\frac{\partial(\theta R)}{\partial t} = -\frac{\partial q_i}{\partial z} \tag{2}$$

Note that for the convenience of calculation, isotopic ratio (R) is used in this study, instead of notation  $\delta$ . The conversion between R and  $\delta$  is:

$$\delta = \frac{R - R_{ref}}{R_{ref}} 1000 \tag{3}$$

where  $R_{ref}$  is the isotopic ratio reference value,  $155.76 \times 10^{-6}$  and  $2,005.2 \times 10^{-6}$  for deuterium and oxygen-18, respectively.

Assuming the topsoil layer has a thickness of  $\Delta z$  and the variation of soil water and isotopic fluxes are uniform within the topsoil layer, then Eqs. (1) and (2) can be linearized as:

$$\frac{\partial \theta}{\partial t} = -\frac{\left(Q - (P + E)\right)}{\Delta z} \tag{4}$$

$$\frac{\partial(\theta R)}{\partial t} = -\frac{\left(Q_i - (P_i + E_i)\right)}{\Delta z} \tag{5}$$

with relationships between water and isotopic fluxes are:

$$\begin{cases}
Q_i = QR \\
P_i = PR_P \\
E_i = ER_E
\end{cases}$$
(6)

105

where R,  $R_P$ , and  $R_E$  are isotopic ratio of soil water in the uppermost layer, precipitation, and evaporation, respectively. Defining the soil water storage (V) of the topsoil layer is  $\theta \Delta z$ , then Eqs. (4) and (5) can be rewritten as:

$$\frac{\partial V}{\partial t} = P + E - Q \tag{7}$$

$$\frac{\partial(VR)}{\partial t} = PR_P + ER_E - QR \tag{8}$$

Combining Eqs. (7) and (8), the E/P ratio can be solved under different assumptions:

#### 90 (1) SS method: Steady state evaporation characterized with constant soil water volume and isotopic ratio

When soil evaporation reaches a steady state, temporal variations in soil water storage and isotopic composition within the uppermost soil layer become negligible. Under these conditions, Eqs. (7) and (8) can be rewritten as:

$$P+E=Q (9)$$

$$PR_P + ER_E = QR \tag{10}$$

Defining the ratio of evaporation to precipitation (E/P) as x and the ratio of Q to P as y, both can be solved analytically from Eqs. (9) and (10):

$$x = \frac{R - R_P}{R_E - R} \tag{11}$$

$$y = \frac{R_E - R_P}{R_F - R} \tag{12}$$

where R and  $R_P$  are measurable,  $R_E$  can be estimated using Craig-Gordon model:

$$R_E = \frac{E_i}{E} \tag{13}$$

where E and  $E_i$  are evaporative water and isotopic fluxes, respectively, based on the vapor concentration between soil surface and atmosphere:

$$E = \frac{cvsat \ RH_{soil} - cvsat \ RH_{atmos}}{\rho} \tag{14}$$

$$E_{i} = \frac{cvsat \, RH_{soil} \, \alpha \, R\text{-}cvsat \, RH_{atmos} \, R_{atmos}}{\alpha_{k} \, \rho} \tag{15}$$

Consequently, Eq. (13) can be rewritten by combining Eqs. (13), (14) and (15):

$$R_E = AR - B \tag{16}$$

with  $A = \frac{RH_{soil} \alpha}{\alpha_k}$ ,  $B = \frac{RH_{atmos} R_{atmos}}{\alpha_k}$ .

In Eqs. (14) and (15), *cvsat* is saturated vapor concentration,  $RH_{soil}$  and  $RH_{atmos}$  are soil and atmospheric relative humidity, respectively; R and  $R_{atmos}$  are isotopic ratio of soil and atmospheric water,  $\alpha$  and  $\alpha_k$  are equilibrium and kinetic fractionation

factors (Fu et al., 2025). Note that the estimated value of x (Eq. 11) should be negative, as the negative sign indicates the direction of evaporation is opposite to that of precipitation (P).

# (2) NSS method: Non-steady state characterized by dynamic soil water volume and isotopic ratio, but caused by evaporation only

Under this framework, Eqs. (7) and (8) can be simplified as:

$$\frac{\partial V}{\partial t} = E \tag{17}$$

$$\frac{\partial(VR)}{\partial t} = ER_E \tag{18}$$

Defining the ratio of final soil water storage (V) to the initial soil water storage ( $V_0$ ) is  $f(f = \frac{V}{V_0})$ . R can be analytically derived from Eqs. (17) and (18) (Derivations can be referred to Appendix A):

$$R = -\frac{B}{1-A} + f^{-(1-A)} \left( R_0 + \frac{B}{1-A} \right) \tag{19}$$

where  $R_0$  is the initial soil water isotopic ratio; A and B are defined in Eq. (16). Note that Eq. (19) is generally written in the following form to estimate remaining water fraction of  $V_0$  after evaporation:

130

115

$$f = \left(\frac{R + \frac{B}{1 - A}}{R_0 + \frac{B}{1 - A}}\right)^{\frac{1}{1 - A}} \tag{20}$$

Then, the evaporative loss fraction of the initial soil water volume  $(f_e)$  can be calculated as:

$$f_e = 1 - f = 1 - \left(\frac{R + \frac{B}{1 - A}}{R_0 + \frac{B}{1 - A}}\right)^{\frac{1}{1 - A}}$$
(21)

Consequently, the ratio of evaporation to precipitation, x, can be written as:

$$x = \frac{V_0 f_e}{P} \tag{22}$$

# (3) ISONEVA: Non-steady state evaporation characterized with dynamic soil water storage and isotopic ratio resulted from evaporative and non-evaporative fluxes

When evaporative and non-evaporative fluxes in the topsoil layer are considered, R can be derived from Eqs. (7) and (8) without simplification (see Appendix A for derivations):

135 
$$R = \frac{R_P - Bx}{1 - Ax + x} + f^{-\frac{1 - Ax + x}{1 + x - y}} \left( R_0 - \frac{R_P - Bx}{1 - Ax + x} \right)$$
 (23)

Solutions of x and y from Eq. (23) are introduced in virtual test section and all parameters in Eq. (23) are already defined.

#### 2.2 Method evaluation

#### Virtual test

The virtual test scenario is adapted from a benchmark case, which is characterized by an unsaturated soil column evaporate under non-isothermal conditions and it has been employed in several hydrological model validation studies (Fu et al., 2025; Zhou et al., 2021). In this study, the boundary conditions of this benchmark were modified: (1) the upper boundary condition was changed from evaporation-only to include both evaporation and precipitation; (2) the lower boundary condition was changed from water supplementation to free drainage. Using the modified setup, soil water and isotope profiles within a 1-meter-deep soil column were simulated over a 100-day period using the MOIST model, whose capability to accurately simulate isotope transport in soil was demonstrated by Fu et al. (2025). The simulated data were then used to assess the accuracy of SS, NSS, and ISONEVA by comparing their estimated E/P ratios with the true values derived from MOIST-simulated evaporation and precipitation fluxes across various temporal and spatial scales.

#### 150 Soil information

160

The simulated soil texture is Yolo light clay (Braud et al., 2005). The relationships between soil water content, pressure head, and unsaturated hydraulic conductivity for this soil type is described using the Brooks-Corey model (Brooks & Corey, 1964).

#### 155 Initial and boundary conditions

The initial condition of soil water content is uniformly distributed with a value of 70% saturated soil water content, while the initial isotope profile is uniformly distributed with a value of 0‰. The air temperature and relative humidity were maintained at 40°C and 0.2, respectively, throughout the simulation (Figure 2). The potential evaporation rate is  $2\times10^{-7}$  m s<sup>-1</sup>. Rainfall is assumed to occur every 5 days, with a flux of  $\epsilon\times3\times10^{-7}$  m s<sup>-1</sup> per event, where  $\epsilon$  is a random number between 0 and 1. The isotopic signature of each rainfall event randomly ranging between -50‰ and -10‰, given by  $-50+\epsilon\times40$  (‰). The lower boundary condition is set as free drainage for both water and isotope transport, implying zero gradients in both soil water potential and soil water isotope compositions at the bottom.

$$T_a = 40$$
°C,  $RH_{atmos} = 0.2$ ,  $\delta_a = -20$ %

Figure 2. Description of the simulated soil column and boundary conditions used in MOIST.  $T_a$ ,  $RH_{atmos}$ , and  $\delta_a$  are atmospheric temperature, atmospheric relative humidity, and atmospheric isotopic compositions of oxygen-18;  $h_c$ ,  $\lambda$ ,  $k_{sat}$ ,  $\theta_{sat}$ , and  $\theta_{res}$  are air entry value, pore size distribution parameter, saturated hydraulic conductivity, saturated soil water content, and residual soil water content;  $\Delta z$  denotes the spatial discretization step.

#### E/P ratio evaluation

MOIST outputs soil water and oxygen-18 profiles on a daily scale over a 100-day simulation period. Then, these simulated data are used in SS (Eq. 11), NSS (Eq. 19), and ISONEVA (Eq. 23) to back-calculate E/P ratio. Additionally, the true E/P ratio can be calculated directly from the simulated evaporation and precipitation fluxes provided by MOIST.

Various temporal intervals (from 5 to 100 days) and five spatial intervals (0.01, 0.05, 0.08, 0.1, and 0.2 m) are considered for E/P estimation. The selected time intervals ensure that at least one rainfall event occurs within each period. For a given time interval, the soil water content and isotopic ratio of the topsoil layer are extracted from MOIST outputs at the initial and final of the interval. For instance, under the first 5-day interval, isotopic compositions and soil water content on Day 1 and Day 5 are used to estimate the E/P ratio over these 5 days.

The spatial intervals are selected to reflect typical soil water isotope sampling depths in field studies, where the thickness of the topsoil generally within 0.2 m (Dubbert et al., 2013; Shokri et al., 2008). Then, MOIST is applied to each of the spatial intervals for simulating soil water, isotope, flux profiles, which are used by SS, NSS, and ISONEVA to estimate E/P (Q/P) ratios reversely.

Note that due to the strong linear correlation between  $\delta^2$ H and  $\delta^{18}$ O, particularly under the idealized conditions simulated by MOIST, they provide redundant rather than complementary information. As a result, they cannot be jointly used to independently constrain both x and y. Therefore,  $\delta^{18}$ O is used as the representative tracer in this virtual test.

Since SS and NSS contain only one unknown, which can be solved directly using output data from MOIST. By contrast, ISONEVA involves two unknowns but only one equation, making it an underdetermined problem that lacks a unique analytical solution. Consequently, we rewrite Eq. (23) as the objective function:

$$F(x, y) = abs \left( \frac{R - \frac{R_P - Bx}{1 - Ax + x}}{R_0 - \frac{R_P - Bx}{1 - Ax + x}} - f^{-\frac{1 - Ax + x}{1 + x - y}} \right)$$
(24)

To optimize Eq. (24), we employ a numerical approach that combines Genetic Algorithm (GA) optimization with Monte Carlo simulation. GA is a stochastic global optimization technique well-suited for exploring complex and non-convex solution spaces, but its random nature can lead to variability in the results. To improve reliability and capture the full range of plausible solutions, we embed the GA within a Monte Carlo framework: each group consists of 500 independent GA runs, and the process is repeated 100 times. The pseudo-posterior distributions of E/P (and Q/P) can be generated, and E/P (Q/P) estimates are reported in the form of mean  $\pm$  standard deviation.

The bounds for variables x (E/P) and y (Q/P) in Eq. (24) are [-20, 0] and [-20, 1], respectively. The negative bound for x reflects the potential opposite direction of evaporation relative to precipitation, while the upper limit of 1 for y represents the scenario where all precipitation infiltrates downward as percolation or root water uptake. A negative lower bound for y indicates the potential upward flux compensation from lower layers to topsoil layer.

Note that Eq. (24) contains an optimization trap: when x approaches 1/(A - 1), Eq. (24) approaches zero. This may cause the solver to converge to 1/(A - 1), even though this value is not necessarily the one we want. To avoid this issue, we added a penalty term to Eq. (24):

$$F(x, y) = abs \left( \frac{R - \frac{R_P - Bx}{1 - Ax + x}}{R_0 - \frac{R_P - Bx}{1 - Ax + x}} - f^{-\frac{1 - Ax + x}{1 + x - y}} \right) + \frac{p_s e^{\left(-\left(x - \frac{1}{A - 1}\right)^2\right)}}{p_w}$$
(25)

where  $p_s$  and  $p_w$  represent the penalty strength (10) and penalty width (1×10<sup>-4</sup>), respectively. This penalty term ensures that when x approaches 1/(A - 1), the penalty becomes stronger, while it remains negligible when x is far from 1/(A - 1), thereby preventing the optimizer from falling into the optimization trap.

#### Field test

#### Site description

The field experiment was conducted on continuously weighted soil lysimeters, situated at the École Polytechnique Fédérale de Lausanne (EPFL), in Switzerland (Nehemy et al., 2021). Lysimeters are exposed to atmospheric conditions and monitored for a period of 43 days after the application of an isotopically labelled irrigation event on the 16 May 2018, ending on the 29 June 2018. One bare lysimeter and one vegetated lysimeter are used to monitor evaporation and evapotranspiration, respectively.

#### 220 Measured data

225

230

Within the vegetated lysimeter, soil water content is measured at four depths (0.25, 0.75, 1.25, and 1.75 m) using frequency domain reflectometry probes (FDR; 5TM Devices Inc., USA), while soil water isotopic compositions are sampled at five depths (0.1, 0.25, 0.5, 0.8, and 1.5 m) and analysed at the Watershed Hydrology Lab at the University of Saskatchewan. To harmonize the spatial scales of these two datasets, we define 0-0.25 m as the topsoil layer. Details about the experiment and sample processing can be referred to Nehemy et al., (2021).

Since evaporation measurements from the neighbour bare lysimeter are only available between 4 June and 29 June, thus, the field validation in this study is conducted in this period. Within this period, the daily evaporation rate (measured by the bare soil lysimeter) ranged from 0.97 to 2.27 mm day<sup>-1</sup> (Figure 3). Three precipitation events (including artificial irrigation) took place on June 10, 14, and 26. The smallest daily input is 69.2 mm day<sup>-1</sup> (on June 10), while the largest input is 193.5 mm day<sup>-1</sup> (on June 26) (Figure 3). The input isotopic signals showed a gradual depletion as the precipitation amount increased (Figure 3).

Under this water input pattern, soil water content in the uppermost layer (0-0.25 m) shows a "rise-decline-rise" trend (Figure 3). Notably, unlike the strong correlation linear trend between hydrogen and oxygen stable isotopes in precipitation, the isotope signals in soil water did not exhibit a fully synchronized or linear pattern of change (Figure 3).

Figure 3. Measured evaporation (panel a), input water (precipitation + irrigation, panel b) and isotope signals (panels c and d), soil water contents (panel e) and isotopic signals (panels f and g) from June 6 to June 29. Note that June 6 is the initial date.

#### 240 E/P estimation

Following the sampling frequency from Nehemy et al. (2021), several time intervals (4, 8, 12, 16, 20, and 24 days) are defined to estimate the E/P ratio for each period, starting from June 4. Meanwhile, actual E/P ratios are calculated using evaporation data from the bare lysimeter, serving as a benchmark for evaluating the performance of SS, NSS, and ISONEVA.

The potential daily evaporation ( $E_p$ ) is assumed to range between 0.1- and 10-mm day<sup>-1</sup>. Consequently, the lower and upper bounds of E/P are set based on the ratio of total  $E_p$  to precipitation during each time interval. To account for the effects of root water uptake and artificial irrigation, the bounds for y are set from 0 to 1. Hydrogen and oxygen stable isotopes are independently used in separate optimization runs, and their estimations are averaged as the final E/P (as well as Q/P) for each time interval (Sprenger et al., 2017).

Additionally, due to the lack of in-situ atmospheric vapor isotope measurements, we adopt reference values of  $\delta^2 H = -140\%$  and  $\delta^{18}O = -20\%$ , which are based on cold trap measurements conducted in Vienna under similar climatic and seasonal conditions (Kurita et al., 2012). The two sites share comparable temperature regimes, humidity, and prevailing atmospheric circulation patterns, which support the appropriateness of this substitution. Note that measurements from Kurita et al. (2012) showed that  $\delta^{18}O$  of atmospheric vapor ranges between -27% and -13%;  $\delta^2H$  ranges between -199% and -94%. Accordingly, a sensitivity test is conducted using the lower and upper bounds of these ranges to assess the suitability of the atmospheric isotopic compositions applied in the ISONEVA framework (see Appendix B).

# 260 ET partition

When assuming the topsoil layer root water uptake flux dominates the non-evaporative flux Q, Q/P can be reasonably interpreted as T/P:

$$\frac{Q}{P} \approx \frac{T}{P} \tag{26}$$

Consequently, E/ET can be estimated by:

$$\frac{E}{ET} = \frac{\frac{E}{P}}{\frac{E}{P} + \frac{Q}{P}} = \frac{x}{x+y}$$
 (27)

Note that the derived E/ET from Eq. (27) represents an upper bound of the true E/ET ratio. This is because Q reflects transpiration fluxes from only the uppermost layer of the soil profile (top 0.25 m in the field test of this study), while total T can include additional contributions from deeper layers. If water uptake occurs below the uppermost layer or the percolation in topsoil layer is nonignorable, then Q < T, resulting in an underestimate of total ET, and thus an overestimate of E/ET. Consequently, unless transpiration from deeper soil layers and percolation are negligible, the computed E/ET using this approach should be interpreted as a conservative upper limit rather than an exact value.

#### Method accuracy

Since SS, NSS, and ISONEVA evaluate E/P ratios that represent the average value of the target period, thus, errors from SS, NSS, and ISONEVA are assessed by mean absolute error (MAE):

$$MAE = abs(EP_{ei} - EP_{mi})$$
 (28)

where  $EP_{ei}$  and  $EP_{mi}$  are estimated and measured (or estimated from MOIST in virtual tests) E/P values; N is the total number of measurements; i is the i<sup>th</sup> measurement.

#### 3 Results

#### 3.1 Comparison of estimated E/P and Q/P ratios between SS, NSS, and ISONEVA from virtual dataset.

Across all spatial and temporal intervals, SS often produces E/P estimates based on SS often deviate markedly from the true values, especially for thicker soil layers and larger time intervals (Figures 4a-4e). This bias arises from its inability to account for soil water storage dynamics (Eq. 11). While NSS considers soil water storage dynamic, it still systematically underestimates E/P (Figures 1a-1e) due to ignoring non-evaporative fluxes (infiltration) and resulted in poor soil water mass balance (Eq. 22).

Figure 4. Comparison of estimated soil water evaporation to precipitation ratios (E/P) using the steady-state (SS), non-steady-state (NSS), and ISONEVA methods across varying topsoil layer thicknesses: (a) 0.01 m, (b) 0.05 m, (c) 0.08 m, (d) 0.1 m, and (e) 0.2 m, under various time intervals in x axis (from 5 to 100 days). Simulated ratios from MOIST are served as true values. Note that E/P estimates are negative because we defined a downward positive direction of fluxes. Additionally, absolute error of E/P estimates for each method and layer thickness are shown in panel f. Numbers above the box in panel f are mean absolute errors.

By contrast, ISONEVA provides the most accurate and stable E/P estimates across all scenarios. Its performance is particularly robust at medium to long time intervals (≥30 days). It is worth noting, however, that at very short intervals (5-10 days), ISONEVA may yield higher errors (particularly for fine layers) and occasionally exceeded those of the SS method (Figure 4a). This is because over short timescales, topsoil water storage varies little, making the steady-state assumption approximately valid and limiting the advantage of ISONEVA. As water storage variations grow with time (larger time intervals), ISONEVA becomes increasingly effective.

These differences are quantitatively summarized in Figure 4f. ISONEVA consistently yields the lowest MAE of E/P estimates, reducing error by over 80% on average compared to SS and NSS methods. Unlike SS, whose error increases monotonically with topsoil layer thickness, NSS and ISONEVA both show a U-shaped response: errors initially decrease with increasing topsoil layer thickness, reaching a minimum around 0.08 m, and then rise again. This U-shaped relationship arises likely because overly thin topsoil layers (e.g., 0.01 m) amplify sensitivity to isotopic fluctuations, leading to large errors. As thickness increases to 0.08 m, errors decrease due to improved signal stability without excessive smoothing. Beyond 0.08 m, further thickening (e.g., 0.2 m) dilutes isotopic gradients, reducing sensitivity and increasing errors again. Thus, 0.08 m topsoil layer thickness represents an optimal trade-off between resolution and robustness.

Figure 5 compares estimated Q/P ratios using the SS and ISONEVA methods against the true values (simulated by MOIST) across a range of topsoil layer thicknesses (0.01, 0.05, 0.08, 0.1, and 0.2 m) as a function of temporal intervals (5 to 100

days). NSS method is excluded from this comparison because it does not account for non-evaporative fluxes and thus cannot provide Q/P estimates.

Figure 5. Estimated Q/P ratios using the steady-state (SS) and ISONEVA methods across different topsoil thicknesses: (a) 0.01 m, (b) 0.05 m, (c) 0.08 m, (d) 0.1 m, and (e) 0.2 m, under various time interval (from 5 to 100 days). Simulated Q/P values from MOIST are used as the reference ("True"). Panel f showed absolute errors of SS and ISONEVA methods across all scenarios. Numbers above the box in panel f are mean absolute errors.

The estimated Q/P values from SS often deviate significantly from true Q/P values, especially at larger topsoil thicknesses and longer time intervals (Figures 5a to 5e). By contrast, ISONEVA consistently produces more accurate Q/P estimates, particularly under 0.01, 0.05, and 0.08 m topsoil layer thickness. At these depths, ISONEVA closely tracks the true Q/P values across the full range of integration intervals, with only minor fluctuations under very short time intervals (e.g., 

Figure 6. Estimated and measured E/P ratios from lysimeter data under different temporal intervals. The shaded pink area represents the uncertainty of ISONEVA estimates. The date is shown on the lower x-axis.

While SS estimates were acceptable at shorter intervals (e.g., 4 and 8 days), their accuracy rapidly dropped with longer intervals (Day 16 and 20), mirroring the trends in virtual simulations. However, the NSS method, which relaxes the steady soil water storage constraint, showed moderate improvements (MAE = 0.10) but still systematically underestimated E/P due to the failure of considering non-evaporative fluxes (e.g., infiltration). By contrast, the ISONEVA method delivered the highest accuracy (MAE = 0.04), closely aligning with measured E/P values throughout the period. This confirms the importance of incorporating both evaporative and non-evaporative fluxes to estimate soil evaporation using field-measured isotope data.

Additionally, cumulative ET from the vegetated lysimeter was 351.25 mm, and cumulative E from the bare lysimeter was 44.25 mm, yielding an observed E/ET ratio of 0.126. Based on the total precipitation input (403.65 mm), ISONEVA estimated an E/P ratio of 0.09, corresponding to an inferred E/ET ratio of 0.103, which slightly underestimated the true ratio—consistent with expectations under vegetated conditions. By comparison, the SS and NSS methods yielded significantly lower E/ET values of 0.026 and 0.04, respectively, substantially underestimating soil water evaporation.

Even in the absence of direct ET measurements, ISONEVA provides a conservative upper bound estimate of E/ET as 0.139, which successfully encompassed the observed value (0.126). By contrast, the upper bound from SS is 0.037 and failed to do so. This further demonstrates the practical utility of ISONEVA in real-world applications, especially where direct ET partitioning is unavailable.

#### 4 Discussion

#### 4.1 ISONEVA improves solution space and avoids potential issues from identifying initial values

ISONEVA is more accurate than SS and NSS methods because it explicitly integrates temporal changes in soil water content and isotopic composition, as well as both evaporative and non-evaporative fluxes, which are ignored by SS and NSS. By introducing non-evaporative fluxes (Q), ISONEVA expands the solution space, allowing estimates to better approach the global optimum and reducing biases caused by oversimplified assumptions. As shown in Figure 7, including Q broadens the region of feasible solutions with lower objective function values (highlighted in yellow), thus enhancing the robustness and accuracy of E/P estimates. Additionally, the contour map illustrates why Q/P estimates from ISONEVA are more sensitive than those of E/P, as reflected in the MAE variations (Figures 4f and 5f). This is primarily because the gradient of the objective function in ISONEVA (Eq. 25) is steeper along the Q/P axis than along the E/P axis within the feasible range (Figure 7). Future studies could incorporate additional constraints, such as energy balance, to further narrow the solution space for Q/P and enhance the stability and reliability of model estimates.

Figure 7. Contour map of Q/P vs. E/P fluxes based on the ISONEVA method generated using MOIST-simulated soil water content and isotope data during the first 5-day interval with a 0.01 m spatial resolution. The red circle marks the solution when non-evaporative flux (Q) is not considered (Q/P = 0, NSS), highlighting a local optimum with a higher objective function value. The yellow band indicates the expanded solution space and lower objective function values achieved when including Q, demonstrating the improved robustness and accuracy of ISONEVA in estimating E/P.

Moreover, ISONEVA avoids the common pitfalls associated with defining initial isotopic values associated with NSS. Many studies determine the initial isotopic composition using the intersection of the evaporation line (EL) with the local meteoric water line (LMWL) when using NSS framework (Benettin et al., 2021; Sprenger et al., 2017), implicitly assuming isotopic homogeneity and purely evaporative processes (Javaux et al., 2016). Heterogeneous mixing, new precipitation inputs, and vapor diffusion often disrupt these assumptions in soils. Importantly, the intersection-derived value does not necessarily represent the actual isotopic composition of the initial soil water storage (Benettin et al., 2018). Consequently, the EL—

LMWL intersection often fails to reflect the true evaporation trajectory, potentially resulting in large initial value errors, up to -50% for  $\delta^2 H$  and -8% for  $\delta^{18}O$  (Benettin et al., 2018). These errors can propagate through evaporation estimates, highlighting a critical limitation of NSS in natural, thus intrinsically heterogeneous, soil systems.

Additionally, the so-called "initial value" refers to the isotopic composition of water in the topsoil layer at a specific point in the solution of the governing partial differential equation (Gonfiantini, 1986). This "initial value" is relative rather than absolute: It does not necessarily correspond to the original isotopic compositions at the physical onset of evaporation. Instead, it marks the beginning of a defined calculation period.

ISONEVA circumvents this issue by redefining the initial value as a relative, temporally resolved parameter corresponding to the specific analysis period rather than an absolute physical starting point. This flexible treatment allows continuous, period-specific evaporation estimates without relying on potentially biased EL-LMWL intersections. Despite the increased computational complexity, ISONEVA offers a more reliable framework for estimating soil evaporation by improving the solution space and eliminating errors associated with initial value determination.

#### 4.2 Practical considerations of ISONEVA for field applications.

Virtual tests confirmed that ISONEVA has greater accuracy in E/P and Q/P estimation under a topsoil layer thickness of 0.08 m with a long temporal interval (> 30 days). Moreover, in the field test, ISONEVA achieved good results even with a thicker topsoil layer (0.25 m) and a shorter time interval (

430

### 4.3 ISONEVA offers a robust and scalable diagnostic for soil evaporation

Partitioning ET into E and T remains a central challenge in ecohydrology, especially in arid and semi-arid ecosystems where E/T ratios fluctuate widely in space and time (Rothfuss et al., 2020; Williams et al., 2004). Accurate E estimation provides critical insights into soil–plant–atmosphere interactions, informing sustainable water management and improving understanding of subsurface water dynamics (Good et al., 2015; Sprenger et al., 2016).

Although ISONEVA-derived E/ET represents an upper bound, this conservative estimate is valuable for identifying whether evaporation dominates under specific conditions. By assuming that all transpiration occurs within the topsoil layer, the true E/ET value will always be lower than or equal to this upper bound. Thus, if measured E/ET exceeds this estimate, it suggests potential errors in model assumptions or flux measurements. This upper bound approach is also useful when transpiration is spatially variable or lacks direct measurements, providing a reliable reference point for hydrological assessments.

Compared to non-isotope-based ET partitioning methods, such as sap flow (Rafi et al., 2019), eddy covariance (EC) (Paul420 Limoges et al., 2020), water-use-efficiency approaches (Yu et al., 2022), and evaporation-to-precipitation complementary
methods (Wu et al., 2024; Zhang and Brutsaert, 2021), ISONEVA offers distinct advantages. Its strength lies in minimal data
requirements, relying primarily on soil water content and isotopic composition, along with basic meteorological variables.
This eliminates the need for detailed vegetation data (e.g., leaf area index, rooting depth) or the extensive calibration datasets
often required by meteorological methods (Table 1; Stoy et al., 2019).

425 Table 1. Summary comparison of ISONEVA with other common ET partitioning approaches in terms of data requirements, ability to directly estimate soil evaporation, vegetation sensitivity, and scalability.

| Approach          | Data requirements | Soil E direct | estimate Vegetation sensiti | vity Scalability |
|-------------------|-------------------|---------------|-----------------------------|------------------|
| ISONEVA           | Moderate to Low   | Yes           | Low                         | High             |
| Sap flow          | High              | No            | High                        | Low              |
| WUE-based         | Moderate to High  | No            | High                        | Moderate         |
| Eddy covariance   | High              | No            | Moderate                    | Moderate         |
| E/P complementary | Moderate to High  | No            | Low                         | Variable         |

Moreover, as a soil-based approach, ISONEVA directly quantifies evaporation by constraining transpiration through soil water and isotope balances. This ensures its robustness, even under conditions of canopy-atmosphere decoupling or strong water stress. Consequently, ISONEVA shows lower sensitivity to transient physiological or atmospheric fluctuations than plant-centric methods coupled with photosynthesis.

Lastly, ISONEVA is inherently scalable. Its simple analytical framework and low data demands make it suitable for long-term, large-scale studies. This contrasts with plot-level sap flow or eddy covariance methods, which face logistical and cost

limitations at larger scales. With advances in in-situ soil isotope monitoring (Beyer et al., 2020; Kühnhammer et al., 2022), regional to global applications of ISONEVA are becoming increasingly possible.

In summary, ISONEVA balances simplicity, robustness, and scalability, making it a powerful tool for soil evaporation and ET partitioning across diverse ecosystems and climates. These capabilities are critical for advancing our understanding of hydrological processes and informing agricultural and ecological water management strategies under changing climatic conditions.

#### **5 Conclusions**

This study introduces a novel isotope-based framework that explicitly incorporates non-evaporative fluxes to improve soil evaporation estimates. Traditional steady-state (SS) methods assume constant water and isotopic conditions, which are rarely met in natural soils, while non-steady-state (NSS) models neglect important non-evaporative processes such as infiltration and transpiration, leading to mass balance errors. By integrating both evaporative and non-evaporative fluxes, the proposed framework improves physical realism and enhances the accuracy and robustness of isotope-based estimates. Both virtual simulations and field tests demonstrate that this approach yields robust and realistic long-term estimates of evaporative and non-evaporative flux ratios relative to precipitation, outperforming traditional isotope-based methods. This enhanced capability provides valuable insights into water flux partitioning and plant water use strategies. Moreover, its minimal data requirements and consistent performance across scales make it especially suitable for regions where direct measurements are scarce. These strengths position ISONEVA as a powerful tool for large-scale, long-term assessments of soil evaporation within the soil–plant–atmosphere continuum. Future research could further extend this framework by integrating remote sensing data or coupling it with hydrological models to improve regional and global evaporation estimates.

## 455 Appendix A. Derivations of NSS and ISONEVA

**NSS** 

$$\frac{\partial V}{\partial t} = E \tag{A1}$$

$$\frac{\partial VR}{\partial t} = ER_E \tag{A2}$$

$$V\frac{\partial R}{\partial t} + R\frac{\partial V}{\partial t} = E(AR - B) \tag{A3}$$

$$V\frac{\partial R}{\partial t} = EAR - EB - ER \tag{A4}$$

$$V\frac{\partial R}{\partial t} = \frac{\partial R}{\partial (lnf)}\frac{\partial V}{\partial t} = -EB + (EA - E)R \tag{A5}$$

$$\frac{\partial R}{\partial (lnf)} + (1-A)R = -B \tag{A6}$$

$$R = -\frac{B}{1-A} + f^{-(1-A)} \left( R_0 + \frac{B}{1-A} \right) \tag{A7}$$

Note that the partial differential equation like:

$$\frac{\partial y}{\partial x} + p(x)y(x) = q(x) \tag{A8}$$

has the analytical solution:

$$y = e^{-\int p(x)dx} \left( \int q(x)e^{\int p(x) dx} dx + constant \right)$$
 (A9)

which is used to derive Eq. A7 from Eq. A6 (also Eq. A17 from Eq. A16 below).

#### **ISONEVA**

$$\frac{\partial V}{\partial t} = P + E - Q \tag{A10}$$

$$\frac{\partial VR}{\partial t} = PR_P + ER_E - QR \tag{A11}$$

$$V\frac{\partial R}{\partial t} + R\frac{\partial V}{\partial t} = PR_P + E(AR - B) - QR \tag{A12}$$

$$V\frac{\partial R}{\partial t} = PR_P + EAR - EB - QR - PR - ER + QR \tag{A13}$$

$$V\frac{\partial R}{\partial t} = \frac{\partial R}{\partial (\ln t)} \frac{\partial V}{\partial t} = PR_P - EB + (EA - P - E)R \tag{A14}$$

$$\frac{\partial R}{\partial (lnf)} + \frac{(E+P-EA)}{P+E-Q}R = \frac{PR_P-EB}{P+E-Q}$$
(A15)

$$\frac{\partial R}{\partial (lnf)} + \frac{(1+x-Ax)}{1+x-y} R = \frac{R_P - Bx}{1+x-y}$$
(A16)

$$R = \frac{R_P - Bx}{1 - Ax + x} + \int_{1 - Ax + x}^{1 - Ax + x} \left( R_\theta - \frac{R_P - Bx}{1 - Ax + x} \right) \tag{A17}$$

#### Appendix B. Sensitivity of SS, NSS, ISONEVA on atmospheric isotopic ratio

In the field validation of this study, we used the average atmospheric isotope values reported by Kurita et al. (2012) as the reference for estimating atmospheric vapor isotopes during evaporation. To enhance the reliability of the results, we employed field test data under a 23-day interval and used the upper and lower bounds of  $\delta^{18}$ O (-27‰ to -13‰) and  $\delta^{2}$ H (-199‰ to -94‰) measured by Kurita et al. (2012) to estimate E/P from the vegetated lysimeter. The results showed that soil

490

evaporation estimates from the SS, NSS, and ISONEVA methods were insensitive to variations in atmospheric isotope values (Table S1), confirming the robustness of our field validation.

The limited sensitivity of SS, NSS, and ISONEVA to  $R_{atmos}$  (or  $\delta_{atmos}$ ) arises from the structure of the equation, where  $R_{atmos}$  only appears in the term B (Eqs. 16, 21, and 23). This term contributes additively and is divided by the kinetic fractionation factor, which dampens its overall influence. Additionally, when the residual water fraction f is close to 1 (i.e., limited evaporation), the output is dominated by the initial water isotope ratio  $R_0$ . Even under stronger evaporation conditions (low f), the exponential weighting still suppresses the impact of B, making the estimated E/P relatively insensitive to variations in atmospheric vapor isotopic composition.

Table B1. Estimated E/P ratios using the SS, NSS, and ISONEVA methods under different atmospheric isotopic compositions, based on field data measured by Nehemy et al. (2021) between June 6 and June 29.  $\delta^{18}O_{atmos}$  and  $\delta^{2}H_{atmos}$  are atmospheric isotopic compositions of oxygen-18 and deuterium, respectively.

|         | $\delta^{18}O_{atmos} = -27\%$ | $\delta^{18}O_{atmos} = -20\%$       | $\delta^{18}O_{atmos} = -13\%$        |
|---------|--------------------------------|--------------------------------------|---------------------------------------|
|         | $\delta^2 H_{atmos} = -199\%$  | $\delta^2 H_{atmos} = \text{-}140\%$ | $\delta^2 H_{atmos} = \textbf{-99}\%$ |
| SS      | -0.04                          | -0.04                                | -0.02                                 |
| NSS     | -0.01                          | -0.01                                | -0.005                                |
| ISONEVA | -0.1                           | -0.09                                | -0.12                                 |

#### 495 Code and data availability

The codes are developed in MATLAB (https://doi.org/10.5281/zenodo.17119369) and distributed under the Creative Commons Attribution 4.0 International license. MOIST model is available from Fu & Si (2023) and the raw dataset of field measurements can be accessed from Nehemy et al. (2021).

#### **Author contributions**

500 Conceptualization: HF, BS, and WZ; Method development: HF and BS; Data collection, simulation, analyzation, and visualization: HF, MG, and HL; Writing and revision: HF, MG, HL, DP, JL, BS, and WZ.

#### **Competing interests**

None

# Acknowledgments

This research was supported by the National Natural Science Foundation of China (42507413), National Key R & D Program of China (2022YFD1500100), Outstanding Youth Fund of Heilongjiang Province (JQ2024D002), China Agriculture Research System of MOF and MARA (CARS04).

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
