# Peer review of "Technical note: Including non-evaporative fluxes enhances the accuracy of isotope-based soil evaporation estimates"

_EGUsphere, 2025_

## Author Comment (AC1)

Reviewer comments are highlighted in boldface and italic. Our responses are in dark blue, while light blue texts are the revisions going to be made.

***2. Figure 3: Please make the font larger. Additionally, the color contrast between model results and observations could be enhanced for clarity.***

Thank you for your comments. The font of Figure 3 has been enlarged.

[Figure]

*Figure 3. Measured evaporation (panel a), input water (precipitation + irrigation, panel b) and isotope signals (panels c and d), soil water contents (panel e, top 25 cm) and isotopic signals (panels f and g) from June 6 to June 29. Note that June 6 is the initial date.*

*Additionally, to enhance the clarity of Figure 6, colors have been revised:*

[Figure]

*Figure 6. Estimated and measured E/P ratios from lysimeter data under different*

*temporal intervals. The shaded pink area represents the uncertainty of ISONEVA estimates. The date is shown on the lower x-axis. The light grey line is the horizontal line at E/P is 0.*

**4. Appendix A: The derivation of ISONEVA are pure equations. Adding explanations would be helpful for readers to understand.**

Thank you for your comments. To help readers follow the derivation, we briefly summarize each step in Appendix A.

NSS section (A1 - A9): "Equation A1 expresses the water balance of the topsoil control volume under evaporation-only conditions, where the change in soil water storage ($\frac{\partial V}{\partial t}$) is equal to the evaporation flux $E$.

Equation A2 represents the corresponding isotope mass balance, where $VR$ is the total mass of isotopes in the control volume and $ER_E$ is the isotopic flux associated with evaporation.

By applying the chain rule, Equation A3 relates the change in soil isotopic ratio ($\frac{\partial R}{\partial t}$) to the water balance ($\frac{\partial V}{\partial t}$) and isotope balance ($\frac{\partial (VR)}{\partial t}$).

Rearranging terms yields Equation (A4), which describes the time evolution of soil isotopic composition as a function of the evaporation rate and the isotopic compositions of soil water and evaporated vapor.

Equation A5 rewrites $v\frac{\partial R}{\partial t}$ in relation to $\frac{\partial (\ln f)}{\partial t}$, where f is the ratio of final to initial soil water storage. Then, Equation A6 shows R as a function of $\ln f$, combining the water-storage change with isotopic enrichment processes.

Solving this first-order linear differential equation leads to Equation A7, which provides the analytical solution for the evolution of $R$.

Equations A8-A9 present the general solution to a linear differential equation of the form $\frac{\partial y}{\partial x} + p(x)y = q(x)$, which is used to derive Equation A7 from Equation A6."

ISONEVA section (A10 - A17): "Equation A10 represents the water mass balance of

the topsoil control volume, where changes in soil water storage $(\frac{\partial V}{\partial t})$ are determined by precipitation ($P$), evaporation ($E$), and percolation ($Q$).

Equation A11 describes the corresponding isotope mass balance, where $VR$ is the total mass of isotopes stored in the control volume. The terms on the right-hand side represent isotopic inputs from precipitation ($PR_P$), isotopic enrichment during evaporation ($ER_E$), and isotopic losses through percolation ($QR$).

To obtain an equation for the evolution of soil water isotopic composition ($R$), Equations A10 and A11 are combined, this leads to Equations A12-A14, which express the temporal evolution of $R$ in terms of water fluxes and their isotopic compositions.

Like NSS derivations, Equation A14 is rewritten in terms of the derivative of $R$ with respect to $ln(f)$, this transformation yields Equation A15.

Finally, Equation A15 can be further simplified to A16, which is a first-order linear differential equation. It can be solved analytically using A9 and results in A17, which is the basis of the ISONEVA estimation."